# DISCO: DISCrete nOise for Conditional Control in Text-to-Image Diffusion Models

**Longquan Dai, Wu Ming, Dejiao Xue, He Wang, and Jinhui Tang**[*]
**Nanjing University of Science and Technology, Nanjing, China**
{dailongquan, wuming, xuedejiao, wanghe, tangjinhui}@njust.edu.cn

## Abstract

A major challenge in using diffusion models is aligning outputs with user-defined conditions. Existing conditional generation methods fall into two major categories: classifier-based guidance, which requires differentiable target models and gradient-based correction; and classifier-free guidance, which embeds conditions directly into the diffusion model but demands expensive joint training and architectural coupling. In this work, we introduce a third paradigm: DISCrete nOise (DISCO) guidance, which replaces the continuous conditional correction term with a finite codebook of discrete noise vectors sampled from a Gaussian prior. Conditional generation is reformulated as a code selection task, and we train prediction network to choose the optimal code given the intermediate diffusion state and the conditioning input. Our approach is differentiability-free, and training-efficient, avoiding the gradient computation and architectural redundancy of prior methods. Empirical results demonstrate that DISCO achieves competitive controllability while substantially reducing resource demands, positioning it as a scalable and effective alternative for conditional diffusion generation. Code is available at https://github.com/dailongquan/disco.

## 1 Introduction

Diffusion models [4, 8] have achieved impressive success across diverse domains [11, 25, 47, 51], scaling to billions of samples [27] and establishing themselves as foundational generative models. They are particularly effective for conditional generation tasks [1, 13, 32, 44, 46] and support a wide range of downstream applications [26, 38, 39, 30, 40, 31]. Consequently, the challenge of conditional generation—tailoring outputs to satisfy user-defined constraints such as labels, attributes, or spatiotemporal cues—is becoming increasingly critical.

Conditional generation in diffusion models typically follows one of two paradigms: classifier-based guidance[10] and classifier-free guidance[15]. Both require training a dedicated model tailored to each conditioning signal. More recently, training-free guidance, a subclass of classifier-based methods, has emerged. These approaches leverage off-the-shelf, differentiable target predictors—such as classifiers, loss functions, or energy functions—to steer generation without additional training. The predictor evaluates sample quality and provides gradients to guide the diffusion process toward desired outputs.

Classifier-based guidance[10] trains a noise-conditional classifier to estimate the class probability $p_\theta(\boldsymbol{c}|\boldsymbol{z}_t)$. Training-free guidance[1, 13, 32, 44, 46] builds on this idea from an energy-based perspective [46], using pretrained target networks—such as depth or segmentation models—to approximate $p_\theta(\boldsymbol{c}|\boldsymbol{z}_t)$ without training a dedicated classifier. While these predictors still require pretraining, their decoupling from the diffusion model reduces overall computational cost. A key limitation, however, is that these approaches require differentiable predictors, as guidance relies on computing the gradient $\nabla_{\boldsymbol{z}_t} \log p(\boldsymbol{z}_t|\boldsymbol{c})$ to apply conditional corrections during sampling.

---

[*]Corresponding author.

39th Conference on Neural Information Processing Systems (NeurIPS 2025).

Classifier-free guidance[15] removes the need for an external classifier by interpolating between the score estimates of a conditional diffusion model and its jointly trained unconditional counterpart to form a conditional correction term. This approach relies on conditioning mechanisms [48] that adapt an unconditional model into a conditional one. A major advantage of classifier-free guidance is its ability to generate outputs more faithfully aligned with the conditioning input [18]. However, this comes at the cost of increased training complexity: the conditional branch modifies features throughout the diffusion model, requiring the full model to be loaded during training. In contrast, classifier-based methods allow the conditional network to be trained independently.

In this paper, we propose a third approach: DISCrete nOise (DISCO) guidance, which overcomes key limitations of existing methods. Both classifier-based and classifier-free methods aim to estimate a conditional correction term $\mathbf{\Delta}(z_t, c)$, which transforms the zero-mean Gaussian noise used in unconditional denoising into a non-zero-mean noise for conditional denoising. Our insight is that, under certain conditions, this non-zero-mean noise can be effectively approximated using a finite codebook of discrete vectors, pre-sampled from a Gaussian distribution. This reframes conditional generation as a code selection problem: given the current diffusion state $z_t$ and conditioning input $c$, the model selects an optimal code from the codebook. To realize this, we introduce a discrete noise prediction network that learns to perform this selection efficiently and effectively. In summary, the contribution of our DISCO are that:

**Differentiability-Free:** Our discrete noise prediction network directly selects the optimal code to substitute for the non-zero-mean Gaussian noise, avoiding the need to compute gradients of the target predictor as required in classifier-based guidance.

**Training Efficiency:** Training the discrete noise prediction network does not require loading or fine-tuning the diffusion model, significantly reducing computational overhead. This is because both inputs—$z_t$ and $c$—as well as the discrete code indices can be precomputed.

**Alignment Improved:** Our method explicitly trains the prediction network to select control noise, providing direct supervision for controllability. In contrast, classifier-based guidance relies on auxiliary classification loss, which does not directly align with the diffusion process.

## 2   Related Work

Classifier-based guidance has primarily focused on training-free approaches, where an independently trained classifier is used to guide the diffusion process during inference without modifying the diffusion model itself. In contrast, classifier-free guidance relies on learned conditioning, typically by integrating an auxiliary conditional network into the diffusion model during training. In this section, we briefly review both categories to contextualize our proposed DISCO method, which introduces a third paradigm for conditional generation that is distinct from both classifier-based and classifier-free approaches.

**Classifier-based guidance** was introduced by Dhariwal and Nichol [10], who trained a time-dependent classifier to approximate the posterior $\log p_t(c|z_t)$ using a pre-trained diffusion model. Later works estimate the gradient $\nabla_{z_t} \log p_t(c|z_t)$ at inference using pre-trained predictors. Chung et al. [6] applied Tweedie's formula to linear inverse problems, which was extended to broader conditional generation by Chung et al. [7], Yu et al. [46], Bansal et al. [1], and Wang et al. [37]. Song et al. [32] reduced bias via multiple samples from a noisy Gaussian prior. Zhu et al. [52] introduced manifold projection, later refined by He et al. [13] with an autoencoder enforcing guidance in the tangent space. Yang et al. [43] further improved this by mitigating manifold drift without strong assumptions. Despite these advances, classifier-based guidance fails when the target predictor is non-differentiable.

**Classifier-free guidance** extends the conditional processing capabilities of diffusion models by modifying the architecture to directly incorporate conditioning information. ControlNet [48] proposes to utilize the trainable copy of the UNet encoder in the diffusion model to encode extra condition signals into latent representations and then apply zero convolution to inject into the backbone of the UNet in diffusion modal. However, ControlNet is a single-modality framework and requires a separate model for each modality. To address this, unified ControlNet-like models [17, 18, 20, 29, 22, 49, 50, 34] are proposed to handle diverse control signals by inject multi-modality into original diffusion model. Despite their flexibility, these models are typically resource-intensive to train.

# 3 Preliminaries

In this section, we delve into two fundamental theorems that theoretically guarantee the practicality of our discrete noise guidance approach.

**Theorem 1.** *Let $n \geq 2$ denote the ambient dimension, and let $\theta = \angle(\boldsymbol{\mu}, \boldsymbol{\epsilon})$ be the angle between the mean vector $\boldsymbol{\mu}$ and a random vector $\boldsymbol{\epsilon} \sim \mathcal{N}(\boldsymbol{\mu}, \boldsymbol{I})$. Then, $\upsilon = \cos^2 \theta = \left( \frac{\langle \boldsymbol{\mu}, \boldsymbol{\epsilon} \rangle}{\|\boldsymbol{\mu}\| \cdot \|\boldsymbol{\epsilon}\|} \right)^2$ follows the non-central beta distribution:*

$$\upsilon \sim Beta\left( \alpha, \ \beta; \ \lambda \right). \tag{1}$$

*where $\alpha = \frac{1}{2}$, $\beta = \frac{n-1}{2}$, and $\lambda = \|\boldsymbol{\mu}\|^2$. Let $_2F_1\left( a, \ b; \ c; \ z \right)$ be a Gaussian hypergeometric function. The expectation of $\cos^2 \theta$ is given by*

$$\mathbb{E}[\upsilon] = \frac{\alpha + \lambda}{\alpha + \beta + \lambda} \cdot {}_2F_1\left( 1, \alpha + 1; \alpha + \beta + 1; \frac{1}{1 + \lambda} \right) \cdot \frac{1}{\alpha + \beta}. \tag{2}$$

This theorem can be leveraged to assess whether a vector $\boldsymbol{\epsilon}$ is statistically consistent with being drawn from $\mathcal{N}(\boldsymbol{\mu}, \boldsymbol{I})$. Specifically, one can evaluate its cumulative probability under the above distribution $p = P(\upsilon \leq \cos^2 \angle(\boldsymbol{\mu}, \boldsymbol{\epsilon}))$, where $\upsilon \sim \text{Beta}\left( \alpha, \ \beta; \ \lambda \right)$. A large $p$-value suggests that the observed angle is likely under the assumed distribution. Thus, we have the following definition:

**Definition 1.** *Under the setup described above, we consider that a vector $\boldsymbol{\epsilon}$ is likely to be sampled from $\mathcal{N}(\boldsymbol{\mu}, I)$ if $\left( \frac{\langle \boldsymbol{\mu}, \boldsymbol{\epsilon} \rangle}{\|\boldsymbol{\mu}\| \cdot \|\boldsymbol{\epsilon}\|} \right)^2 > \mathbb{E}[\upsilon]$. Equivalently, if the angle between $\boldsymbol{\mu}$ and $\boldsymbol{\epsilon}$ satisfies $\angle(\boldsymbol{\mu}, \boldsymbol{\epsilon}) < \arccos\left( \sqrt{\mathbb{E}[\upsilon]} \right)$, we regard $\boldsymbol{\epsilon}$ as being drawn from $\mathcal{N}(\boldsymbol{\mu}, I)$.*

Let $K \geq 1$ be the number of independent random samples. We consider independent random vectors $\mathcal{C}_n^K = \left\{ \boldsymbol{\varepsilon}^{(1)}, \boldsymbol{\varepsilon}^{(2)}, \ldots, \boldsymbol{\varepsilon}^{(K)} \right\}$ sampled from $\mathcal{N}(\boldsymbol{0}, \boldsymbol{I})$ in the $n$-dimensional space, and let $\boldsymbol{\mu} \in \mathbb{R}^n$ be a vector. For each $i = 1, \ldots, K$, $\theta_i = \angle(\boldsymbol{\mu}, \boldsymbol{\epsilon}^{(i)})$ is defiend as the angle between $\boldsymbol{\mu}$ and $\boldsymbol{\epsilon}^i$. Thus $\upsilon_i = \cos^2 \theta_i$ be independent sampled from the non-central beta distribution (1).

The maximum of $\{\upsilon_i\}$ is given by $\nu_K = \max_{1 \leq i \leq K} \upsilon_i$. As $K \to \infty$, the typical maximum of the cosine square similarity $\nu_K$ is a value $b_K$ such that: $\mathbb{P}(\nu_K \approx b_K) \to 1$. When the sample size $K$ becomes large, the typical maximum refers to the value that the maximum is most likely to take. It is not exactly the expected value, nor is it concerned with rare extreme fluctuations; rather, it describes where the maximum tends to be most of the time. We use it to describe how similar the random set $\mathcal{C}_n^K$ is to $\boldsymbol{\mu}$ and have the following theorem to estimate the typical maximum $\nu_{\mathcal{C}_n^K}$.

**Theorem 2.** *Under the setup described above, let $\{\upsilon_i\}_{i=1}^K$ be $K$ samples drawn from the non-central beta distribution $Beta\left( \alpha, \ \beta; \ \lambda \right)$. When $n$ is moderately large and $K$ is not too small, the typical maximum $\nu_{\mathcal{C}_n^K}$ is approximately the same:*

$$\nu_{\mathcal{C}_n^K} \approx \frac{2 \log K}{n} \tag{3}$$

This theorem reveals that in high-dimensional spaces with large $n$, any two vectors are nearly orthogonal as the typical maximum of approaches $1$. To increase the similarity between two vectors, we must decrease the dimension $n$ and increase the number of samples $K$. Since each $\upsilon_i$ measures the cosine square similarity between $\boldsymbol{\varepsilon}^{(K)}$ and $\boldsymbol{\mu}$, the typical maximum $\nu_{\mathcal{C}}$ describes how the set $\mathcal{C}$ can approach $\boldsymbol{\mu}$. Thus, under the setup described above, the definition in the following is reasonable.

**Definition 2.** *If $\nu_{\mathcal{C}_n^K} > 0.15 n \mathbb{E}[\upsilon]$, we say the set $\mathcal{C}_n^K$ can provide a sample for the distribution $\mathcal{N}(\boldsymbol{\mu}, \boldsymbol{I})$ and say $argmin_{\boldsymbol{\epsilon}^{(i)} \in \mathcal{C}_n^K} \cos^2 \angle(\boldsymbol{\mu}, \boldsymbol{\epsilon}^{(i)})$ as the sample.*

# 4 DISCO

In this section, we introduce DISCO, a method for conditional generation using diffusion models. We begin by showing that conditional generation can be achieved by sampling noise from a non-zero-mean Gaussian distribution. Next, we demonstrate that this sampling process can be replaced by selecting discrete codes from a predefined codebook. This insight implies that accurately predicting the correct discrete code based on the input condition enables effective conditional generation. To realize this, we describe how to train a network that predicts these discrete codes from intermediate diffusion states and the conditioning input.

## 4.1 Non-zero-mean Gaussian Distribution for Conditional Generation

To derive conditional diffusion models, Dhariwal and Nichol [10] introduced *classifier guidance*, which modifies the score $\epsilon_\theta(z_t, c) \approx -\sigma_t \nabla_{z_t} \log p(z_t|c)$ by adding the gradient of the log-likelihood from a classifier model $p_\theta(c|z_t)$, which is proportional to $\exp(\mathcal{E}(z_t, c))$, as shown below:

$$\tilde{\epsilon}_\theta(z_t, c) = \epsilon_\theta(z_t, c) - w\sigma_t \nabla_{z_t} \log p_\theta(c|z_t) = \epsilon_\theta(z_t, c) + \underbrace{-w\sigma_t \nabla_{z_t} \log \mathcal{E}(z_t, c)}_{\text{Conditional Correction } \Delta(z_t, c)}, \qquad (4)$$

where $w$ is a parameter controlling the strength of the classifier guidance. The modified score $\tilde{\epsilon}_\theta(z_t, c)$ is then used in place of $\epsilon_\theta(z_t, c)$ when sampling from the diffusion model, resulting in approximate samples from the distribution: $\tilde{p}_\theta(z_t|c) \propto p_\theta(z_t|c)p_\theta(c|z_t)^w$. This approach effectively increases the probability of data where the classifier $p_\theta(c|z_t)$ assigns high likelihood to the correct conditions.

Since classifier guidance relies on gradients from an image classifier, Ho and Salimans [15] aim to remove the need for the classifier and introduce *classifier-free guidance*. This approach modifies $\epsilon_\theta(z_t, c)$ in a way that produces the same effect as classifier guidance but without using a classifier. They train an unconditional denoising diffusion model $p_\theta(z)$, parameterized by $\epsilon_\theta(z_t, c = \varnothing)$, along with the conditional model $p_\theta(z|c)$, parameterized by $\epsilon_\theta(z_t, c)$, where $\varnothing$ denotes the absence of the class identifier $c$. Sampling is then carried out using the following linear combination of the conditional and unconditional score estimates:

$$\tilde{\epsilon}_\theta(z_t, c) = \epsilon_\theta(z_t, c) + \underbrace{w(\epsilon_\theta(z_t, c) - \epsilon_\theta(z_t, \varnothing))}_{\text{Conditional Correction } \Delta(z_t, c)} \qquad (5)$$

The DDPM denoising step [16] for conditional generation thus becomes

$$\begin{aligned}
z_{t-1} &= \frac{1}{\sqrt{\alpha_t}} \left( z_t - \frac{1 - \alpha_t}{\sqrt{1 - \bar{\alpha}_t}} \tilde{\epsilon}_\theta(z_t, c) \right) + \rho_t \epsilon', \quad \epsilon' \sim \mathcal{N}(0, I) \\
&= \frac{1}{\sqrt{\alpha_t}} \left( z_t - \frac{1 - \alpha_t}{\sqrt{1 - \bar{\alpha}_t}} \epsilon_\theta(z_t, c) \right) + \rho_t \epsilon, \quad \epsilon \sim \mathcal{N}(\mu(z_t, c), I)
\end{aligned} \qquad (6)$$

where $\mu(z_t, c) = -\frac{1 - \alpha_t}{\sqrt{\alpha_t}\sqrt{1 - \bar{\alpha}_t}} \Delta(z_t, c)$ and the last term $\epsilon$ is sampled from a non-zero-mean Gaussian distirbution $\mathcal{N}(\mu(z_t, c), I)$.

## 4.2 Discrete Noise for Conditional Generation

Equation (6) shows that in conditional DDPM sampling, noise is drawn from a non-zero-mean Gaussian distribution. We propose replacing this noise with a code selected from a fixed codebook $\mathcal{C}_n^K = \varepsilon^{(1)}, \ldots, \varepsilon^{(K)}$, where each code $\varepsilon^{(k)}$ is independently sampled from $\mathcal{N}(0, I)$ and remains unchanged during training and inference. The core idea is that an appropriate code from this set can serve as an effective discrete approximation of the non-zero-mean Gaussia noise, enabling a simple codebook lookup in place of continuous sampling.

Theorem 1 provides a criterion for selecting a good substitute for Gaussian noise. It shows that the squared cosine similarity between $\mu$ and $\epsilon \sim \mathcal{N}(\mu, I)$ follows a non-central beta distribution. If there exists a code $\epsilon^{(i)} \in \mathcal{C}_n^K$ whose squared cosine similarity with $\mu(z_t, c)$ exceeds the expected value $\mathbb{E}[v]$ of this distribution, then $\epsilon^{(i)}$ can be considered a valid substitute under Definition 1.

Theorem 2 evaluates whether the codebook $\mathcal{C}_n^K$ can provide such a substitute. It defines the typical maximum similarity $\nu_{\mathcal{C}_n^K}$ between $\mu(z_t, c)$ and the codes in $\mathcal{C}_n^K$ as in Equation (3). If $\nu_{\mathcal{C}_n^K} > \mathbb{E}[v]$, the codebook is deemed sufficient to approximate samples from the target non-zero-mean Gaussian distribution $\mathcal{N}(\mu(z_t, c), I)$.

Since the latent space of SD 1.5 has size $4096 = 4 \times 64 \times 64$, the dimensionality of the noise $\epsilon$ in Equation (6) is also 4096. According to Theorem 2, in such high-dimensional spaces, a predefined codebook $\mathcal{C}_n^K$ cannot provide a good approximation to samples from a non-zero-mean Gaussian distribution, as most vectors tend to be nearly orthogonal. To address this, we divide $\epsilon$ into 256 blocks $\{\epsilon_i\}_{i=1}^{256}$, each of size $4 \times 4 \times 4$, so that $\epsilon = \text{cat}(\{\epsilon_i\})$, and each block has dimensionality 64. In this configuration, the expected value of $\text{Beta}(0.5, 31.5; \lambda)$ is 0.0278, with $\lambda = \|\mu_i\|^2 \approx 256$ based on empirical statistics. For a predefined codebook $\mathcal{C}_{64}^{4096}$ with 4096

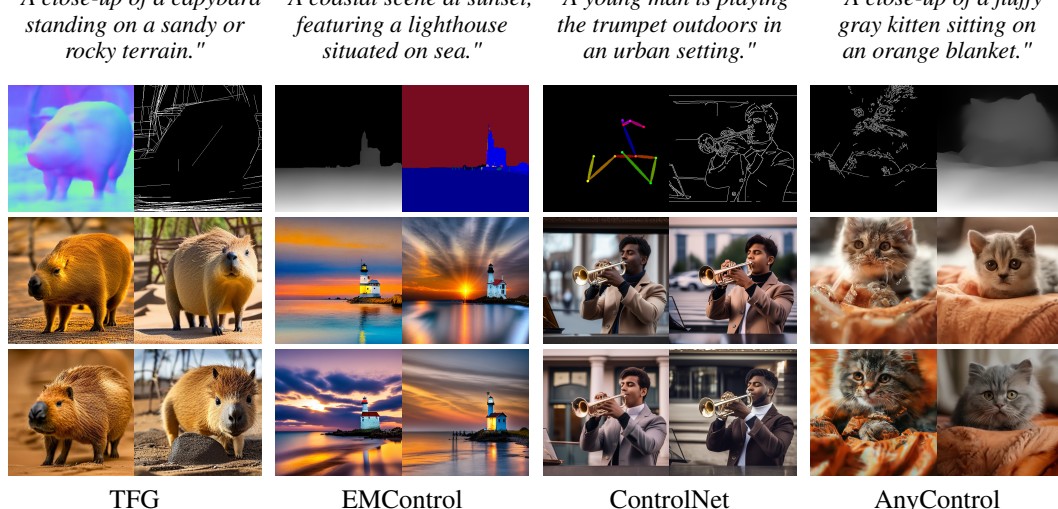

Figure 1: Conditional Generation with Discrete Noise. The first two rows present the text prompt and the corresponding conditioning input. The third row shows the outputs generated using continuous noise. The final row displays the results generated using discrete noise. The results demonstrate that our method maintains generation quality without degradation.

vectors sampled from a 64-dimensional standard Gaussian, the typical maximum squared cosine similarity is 0.263. Since this exceeds the expected similarity from the non-zero-mean distribution, we conclude that $C_{64}^{4096}$ is sufficient to approximate samples from $\mathcal{N}(\boldsymbol{\mu}_i(\boldsymbol{z}_t, \boldsymbol{c}), \boldsymbol{I})$, where $\|\boldsymbol{\mu}_i\| \approx 16$, and $\boldsymbol{\mu}(\boldsymbol{z}_t, \boldsymbol{c}) = \mathrm{cat}(\{\boldsymbol{\mu}_i(\boldsymbol{z}_t, \boldsymbol{c})\})$. This enables us to reformulate Equation (6) as follows:

$$
\begin{aligned}
\boldsymbol{z}_{t-1} &= \frac{1}{\sqrt{\alpha_t}} \left( \boldsymbol{z}_t - \frac{1 - \alpha_t}{\sqrt{1 - \bar{\alpha}_t}} \boldsymbol{\epsilon}_\theta(\boldsymbol{z}_t, \boldsymbol{c}) \right) + \rho_t \mathrm{cat}(\{\boldsymbol{\epsilon}_i\}), \\
\boldsymbol{\epsilon}_i &= \boldsymbol{\varepsilon}^{(k_i)}, \quad k_i = \mathrm{argmax}_{1 \le j \le 4096} \cos^2 \angle(\boldsymbol{\mu}_i(\boldsymbol{z}_t, \boldsymbol{c}), \boldsymbol{\varepsilon}^{(j)})
\end{aligned}
\tag{7}
$$

To validate our approach, we replace the non-zero-mean Gaussian noise used in TFG [44], EMControl [37], ControlNet [48], and AnyControl [34] with our proposed discrete noise. The evaluated algorithms are categorized into two types: classifier-based guidance methods (e.g., TFG and EMControl) and classifier-free guidance methods (e.g., ControlNet and AnyControl). We apply Equations (4) and (5) to compute the conditional correction, followed by Equation (7) to perform conditional generation with discrete noise. The results are presented in Figure 1, where the third row shows outputs generated using the original methods, and the fourth row displays the results with our approach. The comparable visual quality confirms the effectiveness of our substitution strategy.

## 4.3 Conditional Control by Discrete Noise Guidance

Both classifier guidance and classifier-free guidance estimate the condition term $\boldsymbol{\Delta}(\boldsymbol{z}_t, \boldsymbol{c})$ based on the intermediate state $\boldsymbol{z}_t$ and the condition $\boldsymbol{c}$. In the previous section, we discretized the noise $\boldsymbol{\epsilon} \sim \mathcal{N}(\boldsymbol{\mu}(\boldsymbol{z}_t, \boldsymbol{c}), \boldsymbol{I})$ and validated its effectiveness. Building on this, we propose training a network that directly predicts the discrete noise code from $\boldsymbol{c}$ and $\boldsymbol{x}_t$, thereby enabling explicit control over the generation process. Notably, this approach differs fundamentally from prior methods, as it predicts the discrete control noise rather than the condition term $\boldsymbol{\Delta}(\boldsymbol{z}_t, \boldsymbol{c})$ itself.

### 4.3.1 Discrete Noise Predication

In this subsection, we describe how the discrete noise indices $\{k_i\}_{i=1}^{256}$ is predicted from the intermediate features $\boldsymbol{z}_t = \mathrm{cat}(\{\boldsymbol{z}_{t,i}\}_{i=1}^{256})$ and the conditioning input $\boldsymbol{c}$, where each $\boldsymbol{z}_{t,i}$ is a $4 \times 4 \times 4$ block of the intermediate state $\boldsymbol{z}_t$. We adopt the Diffusion Transformer (DiT) [21] as our backbone, which integrates time-step conditioning into a standard encoder-only transformer [36] and employs rotary positional embeddings [33] for spatial encoding.

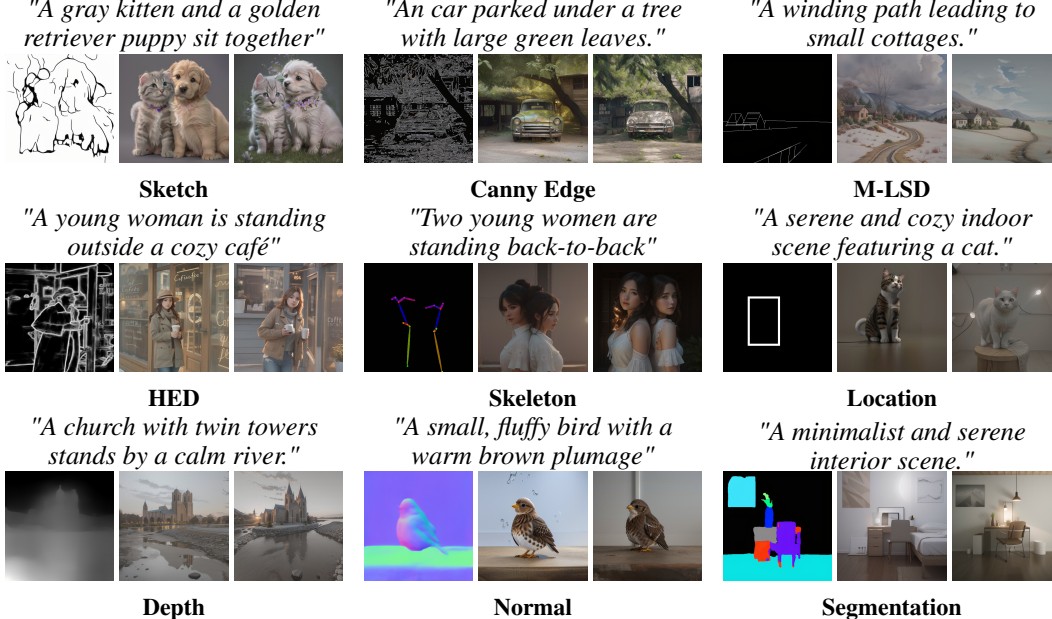

*"A gray kitten and a golden retriever puppy sit together"*    *"An car parked under a tree with large green leaves."*    *"A winding path leading to small cottages."*

**Sketch**    **Canny Edge**    **M-LSD**

*"A young woman is standing outside a cozy café"*    *"Two young women are standing back-to-back"*    *"A serene and cozy indoor scene featuring a cat."*

**HED**    **Skeleton**    **Location**

*"A church with twin towers stands by a calm river."*    *"A small, fluffy bird with a warm brown plumage"*    *"A minimalist and serene interior scene."*

**Depth**    **Normal**    **Segmentation**

Figure 2: Conditional Generation with Diverse Conditions via Discrete Noise. Discrete noise enables seamless integration of various conditions into diffusion models. To demonstrate its versatility, we showcase ten diverse applications: Sketch, Canny Edge, MLSD, HED, Skeleton, Location, Depth, Normal, and Segmentation.

We adopt a BERT-style training scheme [9] for our DiT model. Specifically, we formulate the training as a discrete noise index completion task: a subset of the input discrete noise indices is randomly masked, and the model is trained to recover the missing indices $\{k_i\}_{i\in M}$ using the condition $c$, the intermediate state $z_t$, and the unmasked indices $\{k_i\}_{i\in \overline{M}}$, where $M$ denotes the set of masked positions. Unlike traditional Transformers that operate solely on discrete token sequences, our model uses a hybrid input that combines discrete indices with continuous visual features.

The hybrid inputs to DiT consist of condition $c$, the intermediate state $z_t$, and unmasked discrete noise indices $\{k_i\}_{i\in \overline{M}}$. For each masked index, we replace its original codebook index with a special index $[MASK]$. Conditioned on the unmasked indices, intermediate features $\{z_{t,i}\}_{i=1}^{256}$, and condition $c$, the Transformer is trained to estimate the likelihood of the indices at the masked positions:

$$P(\{k_i\}_{i\in M}|\{k_i\}_{i\in \overline{M}}, \{z_{t,i}\}_{i=1}^{256}, c) = \prod_{i\in M} P(k_i|\{k_i\}_{i\in \overline{M}}, \{z_{t,i}\}_{i=1}^{256}, c) \tag{8}$$

The model is optimized by minimizing the softmax cross-entropy loss between the predicted probabilities and the ground-truth indices. During the inference time, the model starts with the intermediate features $z_t = \text{cat}(\{z_{t,i}\}_{i=1}^{256})$ and the conditioning input $c$ and the discrete noise indices all filled with $[MASK]$. The predicted discrete noise indices are generated by the hybrid-transformer.

### 4.3.2 Training Dataset Preparation

Training the discrete noise prediction network requires data pairs $(z_{t,i}, k_i)$ for a given condition $c$. A straightforward approach is to use intermediate states $z_t = \text{cat}(\{z_{t,i}\}_{i=1}^{256})$ and corresponding discrete noise indices $\{k_i\}_{i=1}^{256}$ generated by an existing conditional method such as FreeDoM [46]. However, this method inherits the limitations of the underlying generation model, potentially compromising data quality. Instead, we adopt a reconstruction-based strategy. Given $c$, we define the energy function in Equation (4) as $\mathcal{E}(z_t, c) = \|z_{0|t} - c\|^2$, where $z_{0|t} = (z_t - \sqrt{1-\bar{\alpha}_t}\epsilon_\theta(z_t, c))/\sqrt{\bar{\alpha}_t}$. This formulation frames conditional denoising as reconstructing $z_0$ from noisy input $z_t$. By applying Equation (7) during reconstruction, we obtain training pairs $(z_{t,i}, k_i)$. Structural cues (e.g., depth or edge maps) extracted from the reconstructed image $z_0$ are then used as conditioning inputs.

Table 1: Quantitative Comparison for Controllable Generation Under Different Conditions. The best results are highlighted in bold. We use FID, CLIP, and CLIP-Ac to evaluate generation quality, and MSE, SSIM, mIoU, and mAP to assess controllability. A dash "–" indicates that the method does not provide a publicly available model for evaluation.

| | Method | Depth | Canny | HED | M-LSD | Segmentation | Normal | Skeleton | Location | Sketch |
|---|---|---|---|---|---|---|---|---|---|---|
| **FID→** | UGD | - | - | - | - | 25.1856 | - | - | - | - |
| | FreeControl | 24.9604 | **17.9107** | 17.4471 | - | 30.8189 | 28.9317 | 34.7962 | - | - |
| | ControlNet | 19.8064 | 18.9211 | **16.1678** | **21.5897** | 20.3947 | 29.5817 | 26.1679 | 29.6951 | 24.0166 |
| | T2I-Adapter | 20.2817 | 24.8351 | - | - | - | - | **24.9437** | - | - |
| | AnyControl | 18.6452 | 19.3719 | - | - | **19.3647** | - | 25.7867 | - | - |
| | DISCO | **18.0431** | 21.3836 | **15.7542** | 22.0951 | 21.1360 | **27.4390** | 25.5083 | **28.8875** | **23.6784** |
| **CLIP↑** | UGD | - | - | - | - | 0.2916 | - | - | - | - |
| | FreeControl | 0.3017 | 0.3214 | **0.3039** | - | 0.3045 | **0.3043** | 0.3016 | - | - |
| | ControlNet | 0.3061 | 0.3085 | 0.3008 | 0.2915 | 0.2997 | 0.2987 | 0.3011 | 0.2917 | 0.3023 |
| | T2I-Adapter | 0.2990 | 0.3045 | - | - | - | - | **0.3111** | - | - |
| | AnyControl | **0.3063** | 0.3032 | - | - | 0.3069 | - | 0.3089 | - | - |
| | DISCO | 0.2952 | **0.3329** | 0.3035 | **0.2951** | **0.3189** | 0.2943 | 0.2920 | **0.2968** | **0.3106** |
| **CLIP-ac↑** | UGD | - | - | - | - | 5.3704 | - | - | - | - |
| | FreeControl | 5.0129 | 5.0010 | 5.0048 | - | 5.0520 | 5.0704 | 5.4050 | - | - |
| | ControlNet | 5.0973 | 5.1213 | 5.1683 | 5.2763 | **5.3920** | 5.2437 | 5.2956 | 5.2737 | 5.0048 |
| | T2I-Adapter | 5.3498 | 5.1650 | - | - | - | - | **5.4802** | - | - |
| | AnyControl | 5.1342 | 5.0485 | - | - | 4.9362 | - | 4.8970 | - | - |
| | DISCO | **5.4225** | **5.4264** | **5.4575** | **5.4341** | 5.3905 | **5.4284** | 5.4794 | **5.3977** | **5.4730** |
| **Controllability** | | MSE↓ | SSIM↑ | SSIM↑ | SSIM↑ | mIoU↑ | MSE↓ | mAP↑ | mAP↑ | SSIM↑ |
| | UGD | - | - | - | - | 0.6843 | - | - | - | - |
| | FreeControl | 99.3874 | 0.4679 | **0.6159** | - | **0.7005** | 0.3569 | 0.5262 | - | - |
| | ControlNet | 87.3678 | 0.4754 | 0.4836 | 0.7447 | 0.4428 | 0.3572 | 0.4332 | **0.3092** | 0.5262 |
| | T2I-Adapter | 89.8904 | 0.4786 | - | - | - | - | 0.5283 | - | - |
| | AnyControl | 88.9634 | **0.5083** | - | - | 0.3382 | - | 0.3764 | - | - |
| | DISCO | **86.6756** | 0.4412 | 0.4752 | **0.7793** | 0.3916 | **0.3443** | 0.4043 | 0.2563 | **0.6118** |

# 5 Experiment

This section presents a comprehensive quantitative and qualitative evaluation of our method, emphasizing its training efficiency and enhanced alignment performance. Additionally, we include an analysis of how dimensionality and codebook size influence the final results.

## 5.1 Training Details & Efficiency

Unlike ControlNet-style methods [34, 45, 48], our discrete noise prediction network does not require loading a frozen Stable Diffusion backbone into GPU memory, which significantly improves training efficiency. Our training process completes in 3 days on a single NVIDIA RTX 3090 (24GB). In contrast, ControlNet [48], IP-Adapter [45], and AnyControl [34] require an A6000 GPU (48GB) and 3+ days for training, according to our experiments.

We train the discrete noise prediction network for 35 epochs with a batch size of 32, using the Adam optimizer with a learning rate of $1 \times 10^{-5}$. During training, both input images and condition maps are resized to $512 \times 512$. We utilize approximately 118,000 images from the COCO2017 dataset [19] across all conditioning types. For human pose estimation, we use a subset of about 6,500 images specifically selected from the "person" category.

## 5.2 Conditional Generation

DISCO effectively integrates diverse conditioning inputs directly into the image generation process of diffusion models, enabling precise and flexible control over outputs. To demonstrate this capability, we present ten single-condition examples in Figure 2, covering a wide range of modalities, including Canny edge [2], MLSD [12], HED [41], Skeleton [3], Location [24], Depth [42], Normal [35], and Segmentation [5]. These examples highlight the versatility and robustness of DISCO in accommodating a broad spectrum of visual conditions.

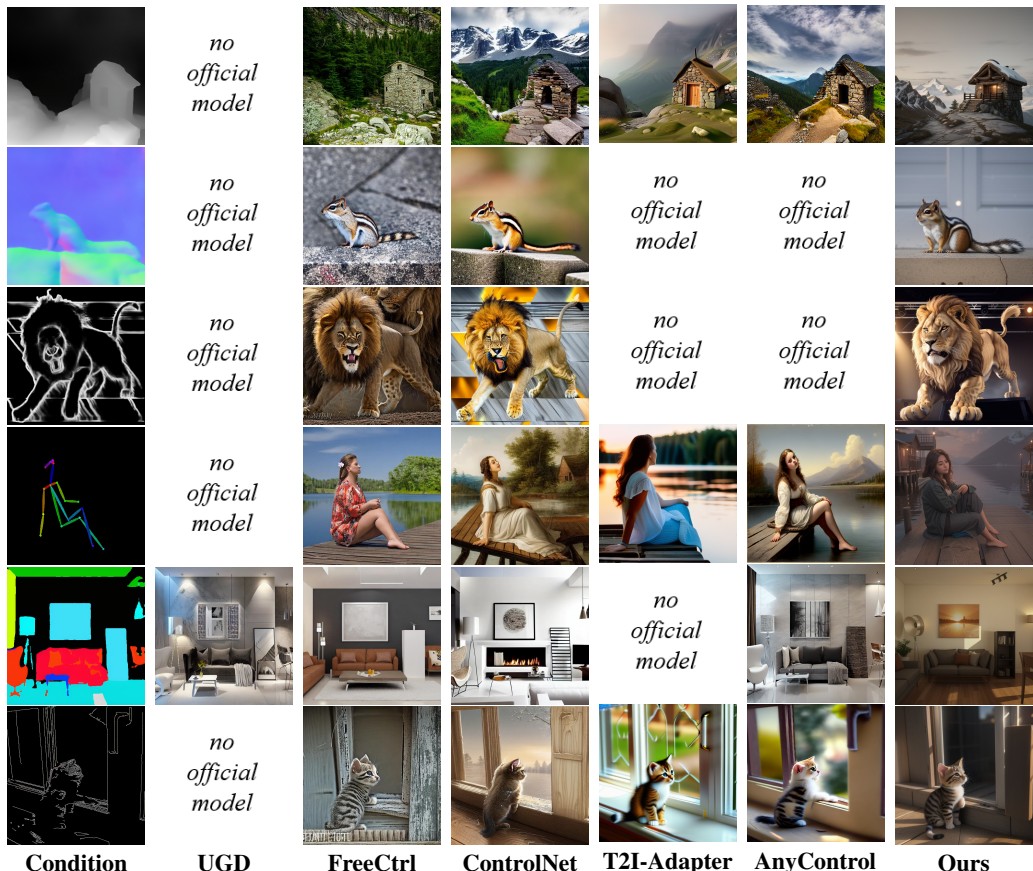

| Condition | UGD | FreeCtrl | ControlNet | T2I-Adapter | AnyControl | Ours |

Figure 3: Qualitative Comparison for Controllable Generation under Different Conditions: We compare our method with five other approaches. Among them, UGD and FreeCtrl belong to the classifier-based guidance category, while ControlNet, T2I-Adapter, and AnyControl are part of the classifier-free guidance category. The results demonstrate that our method is competitive with state-of-the-art techniques.

## 5.3 Qualitative and Quantitative Comparison

In this section, we present both qualitative and quantitative comparisons to demonstrate the controllability of DISCO. It is important to note that not all baseline methods provide publicly available implementations for handling all types of conditions, such as Depth, Canny, HED, M-LSD, Segmentation, Normal, Skeleton, Location, and Sketch. For conditions that are not supported by a given method, we leave the corresponding entries blank.

For a thorough quantitative evaluation, we adopt several metrics, including FID [14], CLIP Score [23], and CLIP Aesthetic Score [28] to assess generation quality. To measure condition controllability, we use SSIM, mAP, MSE, mIoU. As shown in Table 1, our method consistently outperforms existing approaches across various conditions. These results demonstrate that DISCO effectively handles diverse conditioning inputs while producing high-quality, coherent outputs aligned with both textual prompts and constraints. Overall, DISCO achieves superior performance on most metrics, validating its advantage over prior methods.

Figure 3 shows examples of DISCO handling various conditions, demonstrating its ability to generate high-quality results that accurately follow the spatial constraints imposed by the conditioned inputs. The prompts for the six rows are: "a stone hut perched on a rugged, rocky outcrop", "a chipmunk perched on a weathered concrete surface", "a majestic lion with full mane is captured in raw", "a young woman sitting on a wooden dock by the water", and "a small, curious kitten sitting on a wooden windowsill". As illustrated, the generated outputs effectively fulfill the requirements of both the textual prompts and visual conditions.

Table 2: Quantitative comparison with state-of-the-art methods on multiple conditioning modalities. Higher values indicate better performance. DISCO achieves the best results across all conditions.

| | Depth | Canny | HED | MLSD | Seg | Normal | Skeleton | Location | Sketch |
|---|---|---|---|---|---|---|---|---|---|
| UGD | - | - | - | - | 0.12 | - | - | - | - |
| FreeControl | 0.10 | 0.13 | 0.30 | - | 0.11 | 0.34 | 0.11 | - | - |
| ControlNet | 0.21 | 0.28 | 0.33 | 0.42 | 0.23 | 0.27 | 0.14 | 0.37 | 0.41 |
| T2I-Adapter | 0.22 | 0.11 | - | - | - | - | 0.25 | - | - |
| AnyControl | 0.18 | 0.17 | - | - | 0.24 | - | 0.19 | - | - |
| **DISCO (Ours)** | **0.29** | **0.31** | **0.37** | **0.58** | **0.30** | **0.39** | **0.31** | **0.63** | **0.59** |

| Condition | $K = 16$ | $K = 256$ | $K = 1024$ | $K = 4096$ | $K = 65536$ | $K = 131072$ |
|---|---|---|---|---|---|---|

| Condition | $n = 4$ | $n = 16$ | $n = 64$ | $n = 256$ | $n = 1024$ | $n = 2048$ |
|---|---|---|---|---|---|---|

Figure 4: Conditional Generation Quality with Different Dimensionalities $n$ and Codebook Sizes $K$. In the first row, we fix $n = 64$ and vary $K$ to examine its effect on conditional generation, using the prompt "stormtrooper behind a lectern." In the second row, we fix $K = 4096$ and vary $n$, with the prompt "a fluffy white Samoyed dog sitting gracefully on a lush, green lawn."

## 5.4 User Study

In this section, we conducted a user study to further validate the effectiveness of our method, the results of which are summarized in Table 2. For each conditioning type, we randomly selected 50 images. The outputs from different methods were presented to the participants in a random order to avoid positional bias. Participants were then asked to select the best result based on three criteria: *realism*, *consistency with the text prompt*, and *consistency with the input condition*.

Each participant answered three formal questions per condition, plus one validation question, resulting in a total of 28 questions. Responses that failed the validation question were excluded from the subsequent analysis. To compute the final aggregate score, we employed a weighted scoring scheme across the three criteria: realism (30%), text alignment (30%), and condition alignment (40%).

We invited volunteers from our research group, which comprises nearly 100 students, to participate in the study. In total, we collected 76 questionnaires. After filtering out responses that failed the validation check, 72 were considered valid for the final evaluation. As shown in Table 2, our method DISCO achieves the best performance across all nine control conditions, significantly outperforming existing approaches.

## 5.5 Study for Dimensionality and Codebook Size

DISCO is built on the assumption that a predefined codebook $\mathcal{C}_n^K$, constructed from a zero-mean Gaussian distribution, can effectively approximate samples from a non-zero-mean Gaussian distribution. This assumption relies on the existence of at least one code vector in the codebook that is closely aligned in direction with the target mean—i.e., the angle between the selected code vector and the mean of the non-zero-mean Gaussian distribution is small to act as a reliable proxy for conditional generation. Theorem 2 guarantees that this assumption holds when the dimension $n$ is moderately large and the codebook size $K$ is not too small (i.e., $n = 64$, $K = 4096$). Here, we analyze how varying the values of $n$ and $K$ impacts the performance of conditional generation.

In the first row of Figure 4, we fix $n$ and vary $K$. When $K$ is small, the generated outputs fail to align with the conditional input, as the typical maximum of cosine square similarity $\nu_{\mathcal{C}_n^K}$ is close to 0, meaning the codebook cannot approximate samples from the non-zero-mean Gaussian. When $K$ is very large, $\nu_{\mathcal{C}_n^K}$ approaches 1, causing the selected code vector to converge with the mean, reducing randomness in sampling and thereby decreasing output quality. In the second row of Figure 4, we fix $K$ and vary $n$. With small $n$, $\nu_{\mathcal{C}_n^K}$ approaches 1, leading to degraded quality. With large $n$, $\nu_{\mathcal{C}_n^K}$ approaches 0, causing the codebook to fail in aligning with the target mean and making conditional control ineffective. We thus conclude that our method is robust for conditional generation only when $n$ and $K$ are within appropriate ranges, and not at their extreme values.

The configuration ($n = 64 = 4 \times 4 \times 4, K = 4096$) is not the only viable option for discrete noise control. Alternatives such as ($n = 16 = 2 \times 2 \times 4, K = 32$) and ($n = 256 = 8 \times 8 \times 4, K = 65536$) are also feasible, as defined in Definition 2. However, we exclude these due to training efficiency concerns. For example, $n = 16$ requires the discrete noise prediction network to predict 1024 indices, increasing DiT's computational cost by $16\times$ compared to the $n = 64$ setting, which requires only 256 predictions. Conversely, a larger codebook size of $K = 65536$ significantly raises memory and computational demands relative to $K = 4096$. Thus, we adopt ($n = 64, K = 4096$) as a balanced and efficient choice.

## 6 Conclusion

In this paper, we present two key findings: (1) conditional generation in diffusion models can be reformulated as sampling noise from a non-zero-mean Gaussian distribution; and (2) this distribution can be approximated by selecting noise vectors from a predefined codebook, constructed from standard Gaussian samples, which serve as discrete proxies for conditional guidance. Based on this discrete conditional generation perspective, we propose a novel approach—discrete noise prediction—which differs fundamentally from both classifier-based and classifier-free guidance. Specifically, we employ DiT to transform the input ($\boldsymbol{z}_{t,i}, \boldsymbol{c}$) into a noise index $k_i$, indicating the selected entry in the codebook. Experimental results demonstrate the superiority and adaptability of DISCO.

## 7 Acknowledgment

This work was supported by the National Natural Science Foundation of China (62372237,62332010) and the Major Science and Technology Projects in Jiangsu Province under Grant BG2024042.

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
