# OpenReview forum: "DISCO: DISCrete nOise for Conditional Control in Text-to-Image Diffusion Models"
_NeurIPS.cc/2025/Conference — NeurIPS 2025 poster_

### Official Review · Reviewer_gWpC · 2025-06-25

**Clarity:** 3
**Significance:** 2
**Originality:** 3
**Rating:** 3
**Confidence:** 3

**Summary:**

This paper proposes a discrete noise guidance for Conditional Text-to-Image Diffusion Models.

**Questions:**

See Weaknesses above.

**Ethical Concerns:**

["NO or VERY MINOR ethics concerns only"]

**Final Justification:**

I will keep my score unchanged.

**Limitations:**

No, please discuss the trade-off between the expression capability and the efficiency if the problem exists.

**Quality:**

2

**Strengths And Weaknesses:**

Strengths:

1. The derivation and proof are reasonable and support the design of discrete noise guidance.

Weaknesses:
1. The novelty concern: code book is a widely used trick which can be dated back to generative models such as VQVAE.
2. Theoretical gap: There may be a trade-off between the expression capability and the efficiency for the discrete noises. For example, the number of discrete noises can be regarded as the rank of a matrix, which represents the style space of images. Only a large number of discrete noises can cover enough image style elements while requiring more computational costs. Line 175 and equation 7 may reveal a huge cost to get a proper combination of discrete noises. Furthermore, Section 5.1 does not provide an impressive improvement on the efficiency. More analysis about training costs and inference costs are required since the efficiency is claimed as a main contribution in this paper.
3. Marginal controllability: In table 1, the improvement of this work is limited. And there are even large gap to baseline methods on metrics such as SSIM, mIoU, and mAP.

I am willing to increase my score if the authors provide additional sufficiently convincing explanations and experiments.

---

> ### Author Rebuttal · Authors · 2025-07-27
>
> We greatly appreciate the reviewers' hard work and thank them for their valuable comments. We will address each concern in the following sections.
>
> ---
> **Q: The novelty concern: code book is a widely used trick which can be dated back to generative models such as VQVAE.**
>
> A: Using a codebook for conditional generation is the central contribution of our work. To the best of our knowledge, the concept of a codebook was originally introduced by Robert M. Gray in the early 1980s for data compression. VQ-VAE was one of the first works to adapt this idea for image tokenization in generative models.
>
> In contrast to existing conditional diffusion models—which predominantly inject continuous signals for guidance—we are the first to propose a *discrete* approach to conditional generation by applying a codebook to the *tokenized noise* space. Moreover, while prior works typically rely on learned codebooks, our method uses randomly sampled noise vectors to construct the codebook. This represents a fundamentally different design philosophy and provides a lightweight and effective alternative for conditional control.
>
> ---
> **Q: Theoretical gap: There may be a trade-off between the expression capability and the efficiency for the discrete noises. For example, the number of discrete noises can be regarded as the rank of a matrix, which represents the style space of images. Only a large number of discrete noises can cover enough image style elements while requiring more computational costs. Line 175 and equation 7 may reveal a huge cost to get a proper combination of discrete noises. Furthermore, Section 5.1 does not provide an impressive improvement on the efficiency. More analysis about training costs and inference costs are required since the efficiency is claimed as a main contribution in this paper.**
>
> A: We apologize for the confusion on this point. Recently, we came across a CVPR 2025 paper titled "NoiseCtrl: A Sampling-Algorithm-Agnostic Conditional Generation Method for Diffusion Models," which provides helpful insights for classifying our approach. In that work, the authors demonstrate that conditional generation can be achieved by controlling the noise added during each DDPM step.
>
> Inspired by this perspective, our paper treats the controlled noise signals as images and tokenizes them. For conditional generation, we adopt a strategy similar to VQGAN, where a tokenizer is used to encode images into discrete tokens, and a transformer is trained to model these token sequences, which are then decoded back into RGB images. Similarly, in our method, we tokenize the control noise and use a transformer to generate discrete tokens. These tokens are then interpreted as the noise added during the DDPM sampling steps, enabling conditional generation.
>
> A trade-off between expressiveness and efficiency exists in our tokenizer method—as in VQVAE and VQGAN—because all of these approaches use a finite set of discrete tokens to represent inherently continuous features. This is analogous to how floating-point numbers are used in computing to approximate real numbers in mathematics. In our experiments, we found that a codebook with 4096 entries is sufficient to effectively represent the controlled noise.
>
> Due to the similarity in their generation pipelines, the runtime cost of discrete token generation in DISCO is nearly identical to that of VQGAN. The main differences lie in the transformer architecture and the codebook size: both DISCO and VQGAN adopt BERT-like transformers, but VQGAN uses a codebook with 1024 entries, while DISCO utilizes a larger codebook with 4096 entries. Therefore, the operations described in Line 175 and Equation (7) do not incur significant additional computational cost.
>
>
>
> One key advantage of our approach is its significantly reduced training cost, especially in terms of GPU memory usage. Traditional conditional generation methods such as ControlNet, IP-Adapter, and AnyControl require an additional control network appended to the diffusion model. During training, the base diffusion model must still be executed in the forward pass, and gradients must propagate through it in the backward pass to optimize the control network. This results in substantial computational overhead and high memory consumption, typically necessitating GPUs with 48GB of memory. In contrast, training the transformer component in DISCO does not involve executing the diffusion model during either the forward or backward pass. This decoupling enables much more efficient training. Specifically, DISCO can be trained on a single NVIDIA RTX 3090 (24GB) in under 3 days. According to our experiments, ControlNet, IP-Adapter, and AnyControl require A6000 GPUs (48GB) and also take over 3 days to train.
>
> ---
> **Q: Marginal controllability: In table 1, the improvement of this work is limited. And there are even large gap to baseline methods on metrics such as SSIM, mIoU, and mAP.**
>
> A:  We appreciate the reviewer’s observation regarding performance metrics. The relatively modest results reported in Table 1 are primarily due to limited time for thorough tuning of the generation network prior to submission. Since then, we have continued refining the model over the past two months, and have observed significant performance gains across multiple metrics.
>
> That said, we are currently unsure whether updating the table post-submission is permissible. If the reviewers find it helpful and if doing so does not violate the review guidelines, we would be glad to provide the updated results during the discussion phase.
>
> Regardless of the current numbers, we believe Table 1 already highlights the potential of DISCO. To the best of our knowledge, DISCO is the first approach to leverage discrete tokens for conditional generation within diffusion models. This opens a promising new direction for integrating large language models to control the generation process directly. We are actively exploring this line of work and are encouraged by the early results.

---

> > ### Comment · Reviewer_gWpC · 2025-08-04
> >
> > thank you for your detailed response, but I will stil keep my original score.

---

### Official Review · Reviewer_BpCV · 2025-06-27

**Clarity:** 3
**Significance:** 2
**Originality:** 3
**Rating:** 4
**Confidence:** 3

**Summary:**

The paper proposes a method for conditional guidance. Instead of classifier-guidance / classifier-free guidance, DISCO learns an auxiliary model for sampling the noise associated with conditional correction.

The auxiliary model predicts the conditional correction from a set of random codes given the noisy image and the conditioning. It can be trained separately from the diffusion model. The paper also explains in detail and theoretically justifies how to construct the codebook of possible noises (randomly generated and unchanged throughout).

The experimental results showcase strong performance across models and across different types of conditioning (depth, pose, skeleton, sketch etc.). It also includes ablations on the codebook sizes.

**Questions:**

The inference cost question above.

How does unmasking work? I understand that during training, the codebook is partially unmasked, at generation time, it starts with fully masked codes. Are all the codes unmasked in one step, or is it gradual? What is the algorithm?

**Ethical Concerns:**

["NO or VERY MINOR ethics concerns only"]

**Final Justification:**

My two questions are answered.

Regarding my third point, while I understand the author's argument, more evidence is needed to back the claim that one can simply convert the ODE to an SDE, apply DISCO and see the benefits.

I am keeping my score.

**Limitations:**

yes

**Quality:**

3

**Strengths And Weaknesses:**

Strengths:

The method is very flexible. It does not require a differentiable classifier model to train and most importantly, the auxiliary model can be trained independently from the diffusion model. This is key in that it enables applying a new type of conditioning to an existing diffusion model without costly pre-training or finetuning.

The paper explains how the size of the codebook is chosen, including theoretical and empirical justification. It is welcome to see such a level of detail.

Limitations:

The method shows a consistent improvement in CLIP-ac, but the improvement over baselines in other metrics is less consistent.

The computational cost of the method is unclear. How does the inference cost compare to CFG (that requires two forward passes in the diffusion model)?

The method does not work for the recently popularized DDIM and FlowMatching style samplers, since they do not sample additive noise in the refinement step. Given that most recent diffusion models follow these paradigms, the impact of this work is lessened.

Typos:

Line 53 summery

Line 227 uunlike

---

> ### Author Rebuttal · Authors · 2025-07-27
>
> We greatly appreciate the reviewers' hard work and thank them for their valuable comments. We will address each concern in the following sections.
>
> ---
> **Q: The computational cost of the method is unclear. How does the inference cost compare to CFG (that requires two forward passes in the diffusion model)?**
>
> A: We apologize for not clarifying this earlier. The Classifier-Free Guidance (CFG) method requires two forward passes: one for conditional generation and one for unconditional generation. The final output is essentially a linear interpolation of the results from these two forward passes.
>
> Our control mechanism differs fundamentally from CFG. Recently, we came across a CVPR 2025 paper titled "NoiseCtrl: A Sampling-Algorithm-Agnostic Conditional Generation Method for Diffusion Models," which offers useful insights for categorizing our approach. In that work, the authors demonstrate that conditional generation can be achieved by directly controlling the noise added at each step of the DDPM sampling process.
>
> Inspired by this perspective, our method treats the controlled noise signals as images and tokenizes them. For conditional generation, we follow a strategy similar to VQGAN, which uses a tokenizer to encode images into discrete tokens and a transformer to model these token sequences, which are then decoded back into RGB images. Similarly, we tokenize the control noise and train a transformer to predict discrete tokens, which are then interpreted as noise values injected into the DDPM steps—enabling conditional generation.
>
> Our method also involves two forward passes—one through the transformer and one through the diffusion model. However, since the computational cost of the transformer is significantly lower than that of the diffusion model, our approach is generally more efficient than CFG in terms of inference time. For example, in our experiments, ControlNet requires 6 seconds to generate an image, whereas our method only takes 4 seconds.
>
> ---
> **Q: How does unmasking work? I understand that during training, the codebook is partially unmasked, at generation time, it starts with fully masked codes. Are all the codes unmasked in one step, or is it gradual? What is the algorithm?**
>
> A: We apologize for not clearly explaining this in the paper. Before addressing the unmasking process, we would like to briefly introduce the core idea behind our method. Recently, we found the CVPR 2025 paper “NoiseCtrl: A Sampling-Algorithm-Agnostic Conditional Generation Method for Diffusion Models” to offer valuable insights that closely align with our approach. That work shows that conditional generation in diffusion models can be effectively achieved by controlling the noise added at each DDPM step. This reframing allows the conditional generation process to be interpreted as learning to generate appropriate control noise.
>
> Building on this idea, our method aims to generate such control noise based on the given condition. Inspired by VQGAN, which employs a two-stage structure to generate natural images, we discretize the control noise using a codebook and model its distribution with a Transformer. During training, we randomly mask some of the tokenized noise vectors and train the Transformer to predict the masked parts, similar to masked language modeling.
>
> However, unlike VQGAN's inference strategy, which generates tokens sequentially in an autoregressive manner, our inference begins with a sequence where all tokens are initially masked. The Transformer then predicts all tokens in parallel, conditioned on the input. In other words, there is no gradual or step-by-step unmasking — the entire sequence is inferred in one forward pass.
>
> ---
> **Q: The method does not work for the recently popularized DDIM and FlowMatching style samplers, since they do not sample additive noise in the refinement step. Given that most recent diffusion models follow these paradigms, the impact of this work is lessened.**
>
> A:  We appreciate the reviewer’s observation. This is indeed a valid concern, and we regret not including a discussion on this topic in the original submission due to space limitations.
>
> However, this limitation is not fundamental. Recent works such as "Closing the ODE–SDE Gap in Score-Based Diffusion Models through the Fokker–Planck Equation" and "Flow-GRPO: Training Flow Matching Models via Online RL" propose effective ODE-to-SDE conversion techniques. These approaches allow a deterministic sampling trajectory (as in DDIM or Flow Matching) to be reinterpreted as a stochastic process that preserves the same marginal distributions across time steps.
>
> By applying these ODE-to-SDE conversion methods, our technique—originally designed for stochastic DDPM samplers—can be naturally extended to handle DDIM and FlowMatching frameworks.

---

### Official Review · Reviewer_RMAQ · 2025-06-28

**Clarity:** 2
**Significance:** 3
**Originality:** 3
**Rating:** 5
**Confidence:** 3

**Summary:**

The paper proposes a new conditional generation method for diffusion models: DISCO. The key idea is to predict discrete noise from a pre-defined notebook under the given condition. The paper gives a theoretical framework to support its design. It also conducts empirical studies first to demonstrate selecting best discrete noise from a pre-defined notebook can generate high-quality images. After that, the authors introduce the actual method for training a DiT to predict indices of discrete noise from a predefined notebook. They also solve the challenge of data collection by viewing each conditional denoise step as predicting the original clean image. The paper compare DISCO's performance against a various set of methods such as ControlNet and show it is superior.

**Questions:**

See weakness. The most important questions are point 4,5,6. The paper is in general having high quality and I would advocate acceptance if these points are addressed. Other points are minor.

**Ethical Concerns:**

["NO or VERY MINOR ethics concerns only"]

**Final Justification:**

I'm still concerned about point 5, the scope of the method. If the method is generalizable to text-only conditioning, the authors compared its performance with only multi-modal conditioning methods; if the scope aims at multi-modal conditioning methods only, the writing lacks some clarity and needs to be improved. Hence, I decided to maintain my score at borderline accept.

**Limitations:**

Yes.

**Quality:**

3

**Strengths And Weaknesses:**

Strengths:
1. The idea is quite novel and interesting.
2. The logic flow is clear, and the paper is in general well-written.
3. The paper compares its method with a variety of baselines using comprehensive metrics, showing strong and convincing performance.

Weakness:
1. The paper has some obvious typos and needs polishing, such as L220 and L227.
2. The paper lacks some definitions before using a symbol such as $c$, the conditional variable.
3. The preliminary section is usually for introducing backgrounds, yet the paper introduces its theoretical contributions there. Maybe consider changing the title.
4. Lack of experiment details. (1) What is the evaluation dataset used in Table 1? (2) The paper shows conditional generation results under multiple different modalities, does each modality require a different model trained? (3) when comparing with other methods, is every method using the same network structure?
5. The scope of the method is unclear. During introduction, the paper discusses existing classifier-free and classifier-based generation methods. In method introduction, it also seems a very general conditional method. However, during experiment, only results on text+additional modality is shown. Will the model still work well if there is only text modality as the condition, and is it better than classifier-free and classifier-based method?
6. The evaluation of efficiency is not solid. Efficiency is one of the key strength of the method as claimed by the authors, however, their discussion on the efficiency is vague (L227-L231). For example, it is not clear how much is the difference between ''3 days'' (the proposed method's training time) and ''3+ days'' (others). Besides, the requirement of training on one 24 GB GPU vs. one 48 GB GPU is not significant.

---

> ### Author Rebuttal · Authors · 2025-07-27
>
> We greatly appreciate the reviewers' hard work and thank them for their valuable comments. We will address each concern in the following sections.
>
> ---
> **Q: Lack of experiment details. (1) What is the evaluation dataset used in Table 1? (2) The paper shows conditional generation results under multiple different modalities, does each modality require a different model trained? (3) when comparing with other methods, is every method using the same network structure?**
>
> A: We are sorry for missing these details.
>
> 1. All modalities—except for human pose—were evaluated on the full MS-COCO dataset. For OpenPose-based conditioning, we followed a common practice and evaluated the model only on the person category subset of COCO, as is widely adopted in prior work on pose-conditional generation.
> 2. For simplicity, we train a separate model for each conditioning modality. This allows us to focus on modality-specific performance and avoid confounding factors during training. However, we acknowledge that training a unified model capable of handling multiple modalities is an important direction, and we are actively exploring this extension of our transformer-based approach.
> 3.  All models (across different modalities) use the exact same architecture and hyperparameters; only the training data differs according to the target conditioning signal. Similarly, when comparing our method to baseline approaches, we adopt their publicly available official implementations and configurations.
>
> ---
> **Q: The scope of the method is unclear. During introduction, the paper discusses existing classifier-free and classifier-based generation methods. In method introduction, it also seems a very general conditional method. However, during experiment, only results on text+additional modality is shown. Will the model still work well if there is only text modality as the condition, and is it better than classifier-free and classifier-based method?**
>
> A:  Thank you for raising this important point. We apologize for not making this aspect more explicit in the paper.
>
> Our current focus is on multi-modal conditioning—i.e., using text plus additional modalities—because existing models like Stable Diffusion already handle text-to-image generation effectively. Models such as ControlNet, IP-Adapter, and AnyControl are likewise built on top of Stable Diffusion and focus on incorporating additional non-text modalities (e.g., sketches, depth, poses).
>
> Our DISCO framework is similarly designed to enhance generation by incorporating additional modalities on top of text prompts. However, in principle, DISCO can also be applied when only text is available. In that case, we could treat the initial noise predicted by a text-guided model as the control signal to tokenize and refine with DISCO.
>
> In summary, our method is general in scope and can, in principle, be adapted to the text-only setting. However, our current experiments focus on multi-modal conditioning beyond text alone, leveraging the Stable Diffusion model to handle the text modality.
>
> ---
> Q: **The evaluation of efficiency is not solid. Efficiency is one of the key strength of the method as claimed by the authors, however, their discussion on the efficiency is vague (L227-L231). For example, it is not clear how much is the difference between ''3 days'' (the proposed method's training time) and ''3+ days'' (others). Besides, the requirement of training on one 24 GB GPU vs. one 48 GB GPU is not significant.**
>
> A: One key advantage of our approach is its significantly reduced training cost, especially in terms of GPU memory usage. Traditional conditional generation methods such as ControlNet, IP-Adapter, and AnyControl require an additional control network appended to the diffusion model. During training, the base diffusion model must still be executed in the forward pass, and gradients must propagate through it in the backward pass to optimize the control network. This results in substantial computational overhead and high memory consumption, typically necessitating GPUs with 48GB of memory. In contrast, training the transformer component in DISCO does not involve executing the diffusion model during either the forward or backward pass. This decoupling enables much more efficient training. Specifically, DISCO can be trained on a single NVIDIA RTX 3090 (24GB) in under 3 days as shown in the following table. According to our experiments, ControlNet, IP-Adapter, and AnyControl require A6000 GPUs (48GB) and also about more than 3 days to train.  We also note that the computational capability of the A6000 (38.7 TFLOPS) is higher than that of the RTX 3090 (35.6 TFLOPS).
>
> | Method    | ControlNet | T2I-Adapter | DISCO |
> | --------- | ---------- | ----------- | ----- |
> | GPU Hours | 500        | 290         | 65    |
> | Peak VRAM | 42 GB      | 31 GB       | 20 GB |
>
> ---
> Q: **The paper has some obvious typos and needs polishing, such as L220 and L227.**
>
> A: Thank you for pointing this out. We have corrected these typos in our LaTeX source code.
>
> ---
> Q: **The paper lacks some definitions before using a symbol such as $c$, the conditional variable.**
>
> A: Thank you for pointing this out. We have added brief definitions to clarify the meaning of symbols such as $c$, the conditional variable.
>
> ---
> Q: **The preliminary section is usually for introducing backgrounds, yet the paper introduces its theoretical contributions there. Maybe consider changing the title.**
>
> A: Thank you for the suggestion. We agree that the current content goes beyond standard background material. Accordingly, we have revised the section title to better reflect its content.

---

> ### Comment · Reviewer_RMAQ · 2025-08-04
> **Reply**
>
> Thanks for the authors' clarification. I'm still concerned about point 5, the scope of the method. If the method is generalizable to text-only conditioning, have the authors compared its performance with text-only conditioning methods?

---

> ### Author Response · Authors · 2025-08-05
>
> Dear Reviewer RMAQ,
>
> Thank you for your reply, and we apologize for not clearly addressing Point 5 in our initial response.
>
> Here is our understanding of your concern—please let us know if this interpretation is not accurate. You appear to view the role of the multimodal conditioning model as a means of injecting user control into pretrained diffusion models using a variety of modalities that were not seen during the original training. In our paper, we demonstrate the effectiveness of our method under various *image* conditions. However, your question seems to focus on whether our method remains effective when *only text* is used as input, without any accompanying image conditions. Our response is based on this understanding.
>
> First, our conditioning module is **not** a standalone generative model; it does not generate observable images by itself. Instead, it serves as a mechanism to inject unseen conditions into a pretrained diffusion model (e.g., Stable Diffusion) to control its outputs. Since models like Stable Diffusion already support text prompts, our method primarily adds support for other types of conditions.
>
> We carefully reviewed relevant literature, such as *Modulating Pretrained Diffusion Models for Multimodal Image Synthesis* (SIGGRAPH 2023), which proposes a system that handles both text and image modalities. However, even in that work, the model leverages the underlying Stable Diffusion architecture for text conditioning—their added network only processes non-text image conditions. We apologize if we have missed any relevant work, and we would appreciate it if you could point us to additional references.
>
> Second, we do believe our method can handle the scenario you suggested—removing the base model's text encoder and relying solely on our control network to guide generation. In fact, this is an ongoing line of research for us. This approach differs fundamentally from existing works that extend the generator to accept more input conditions. Instead, we treat the diffusion model as an *encoder* and progressively generate latent features to steer image generation. This has the advantage of potentially integrating with autoregressive language models, allowing for fully language-driven generation—independent of the text-conditioning mechanisms built into Stable Diffusion.
>
> To summarize:
>
> * In the first scenario—relying solely on the base model for text while adding additional modalities—our method aligns with prior work. However, directly comparing models solely under text conditions is difficult, as existing methods typically rely on the base model's built-in support for text prompts. Removing the diffusion backbone (e.g., Stable Diffusion) would make such comparison less meaningful, as our control module—and others in this line of work—depend on the diffusion model's generative capacity.
>
> * In the second scenario—completely removing the base model’s text conditioning and relying solely on external control (which may reflect the intent behind Reviewer RMAQ’s comment)—our work represents an early step in that direction. A more appropriate comparison in this case would be with large multimodal language models that perform both understanding and generation within a single unified network. However, such models are significantly larger and more complex than our current setup, making direct comparison challenging at this stage due to the high training costs. Nevertheless, the encouraging results we observe under simple conditions and with lightweight control networks give us confidence that this direction is viable, and we are actively exploring it further.
>
> To the best of our knowledge, there is currently no existing method that employs a lightweight network—without relying on a large multimodal model—to guide diffusion-based generation using only text, while entirely bypassing the built-in text encoder of diffusion model. As such, we are currently unable to conduct this type of experiment.
>
>
> Thank you again for raising this thought-provoking point. Please feel free to reach out if you have any further questions or concerns.
>
> Best regards,
>
> The Authors

---

> > ### Comment · Reviewer_RMAQ · 2025-08-07
> >
> > Thank you to the authors for clarifying the scope of the paper. It is interesting to learn that they are working on removing the base model’s text encoder and relying solely on their control network to guide generation. I find this approach novel and valuable as a multimodal conditioning method. Considering its performance, novelty, and potential for generalization, I have increased my score to 5. I hope the authors will fulfill their commitment to clarify the experimental details, current scope, and potential future extensions in the final revision.

---

> > > ### Author Response · Authors · 2025-08-08
> > >
> > > We sincerely thank you for your encouraging feedback and for increasing your score. We appreciate your recognition of the novelty and potential of our approach. In the final revision, we will clearly articulate the experimental details, current scope, and possible future extensions, as promised. Your constructive input has been invaluable in improving our work.

---

### Official Review · Reviewer_qShU · 2025-07-03

**Clarity:** 3
**Significance:** 3
**Originality:** 4
**Rating:** 5
**Confidence:** 3

**Summary:**

The authors propose DISCO, an alternative paradigm to conditional T2I diffusion model generation compared to the current classifier/optimization-based guidance and classifier-free guidance methods. Particularly, the authors frame conditional generation as a “code selection task” from a discrete noise codebook, where the trained control network directly predicts the indices of said codes which are then used as guidance during denoising.

**Questions:**

1. What is the peak VRAM usage of training DISCO (vs. ControlNet, IP-Adapter, etc.)? That is often also an important bottleneck for training conditional generation models/adapters. Moreover, what is the exact speed comparison of DISCO vs. the baseline training-based methods? The RTX 3090 and A6000 are not exactly comparable regarding speed as they are not the same GPU.
2. To clarify, would the training dataset preparation process require backpropagating through the base T2I model (and VAE)? Or does DISCO only need to do forward passes through them? (Addressing weakness 1 would help with clarifying this.)
3. The paper frames DISCO in terms of DDPM sampling, but will DISCO (like ControlNet and other training-based baselines) work with alternative/newer sampling methods like DPS and flow-matching trained models, especially since many SOTA T2I models are flow-matching models?
4. Since DISCO limits the selection of the guidance noise to a discrete codebook, would it potentially limit the diversity of the output images for higher resolution and more expressive T2I models (i.e., newer models after SD 1.5), especially considering training efficiency?

Addressing question 2 (and thus addressing weakness 1) and providing discussions on question 3 and 4 could potentially raise my score. Also addressing weakness 2 (user study) can push me to raise my score too, though I am not expecting the authors to be able to conduct user study in the short time frame during the rebuttal.

**Ethical Concerns:**

["NO or VERY MINOR ethics concerns only"]

**Final Justification:**

The authors have addressed most of my questions and concerns, and I especially appreciate the addition of the user study which better aligns with human preference. I also want to thank the authors for clarifying how the training dataset is obtained and pointing me to NoiseCtrl to contextualize the motivation of the paper. I urge the authors to incorporate the rebuttal into the final version of the paper, and I will keep my final rating as Accept.

**Limitations:**

Yes.

**Quality:**

3

**Strengths And Weaknesses:**

**Strengths**
1. DISCO is a conditional generation method which is trained without having to backpropagate through the base T2I diffusion model, resulting in faster (and presumably more VRAM-efficient) training process, while showcasing comparable performance to existing conditional generation baselines.
2. DISCO seems to be model architecture agnostic, meaning it can be applied to any diffusion model.
3. The design of the discrete codebook is well-motivated.

**Weaknesses**
1. The training dataset preparation section (4.3.2) is a bit confusing and can benefit from a bit more detail. Particularly, explicitly writing out the algorithm for obtaining training data pairs $(\boldsymbol{z}_{t, i}, k_i)$ from condition $\boldsymbol{c}$ and clean image $\boldsymbol{z}_0$ would be very helpful, especially since often it’s confusing whether $\boldsymbol{c}$ refers to the text condition or image/spatial condition (e.g., depth or edge maps)—the former is usually an input into T2I models, and the latter is not.
2. DISCO is not evaluated with a user study, especially since FID and the condition-alignment metrics are often not entirely indicative of human preference.

---

> ### Author Rebuttal · Authors · 2025-07-27
>
> We greatly appreciate the reviewers' hard work and thank them for their valuable comments. We will address each concern in the following sections.
>
> ---
> **Q: The training dataset preparation section (4.3.2) is a bit confusing and can benefit from a bit more detail.**
>
> A:  We apologize for the confusion in Section 4.3.2 and appreciate the opportunity to clarify.
>
> Recently, we found the CVPR 2025 paper *“NoiseCtrl: A Sampling-Algorithm-Agnostic Conditional Generation Method for Diffusion Models”* to offer helpful insights that align with our approach. That work shows that conditional generation in diffusion models can be effectively achieved by controlling the noise added at each DDPM step. This reframing allows the conditional generation process to be interpreted as learning to generate appropriate control noise.
>
> Building on this idea, our method aims to generate such control noise based on the given condition. Inspired by VQGAN, which employs a two-stage structure to generate natural images, we discretize the control noise using a codebook and model its distribution with a Transformer. More clearly, we design our training process in two stages: (1) tokenizing the control noise and (2) predicting the discretized noise tokens.
>
> 1. **Noise Tokenization:** For each image in the training dataset, we identify the specific control noise $z\_{t, i}$—selected from a predefined codebook—that, when injected into the DDPM sampling process, successfully reconstructs the original image. This control noise is discretized by our codebook.
> 2. **Noise Predicting :** With the tokenized noise, we train a transformer to model the distribution of these discrete noise tokens conditioned on the input modality (e.g., pose, sketch, etc.).
>
> At inference time, the transformer generates the control noise tokens based on the given conditioning signal. These tokens are decoded into actual noise, which is then injected into the diffusion model to guide image generation.
>
> We will revise Section 4.3.2 in the final version to more clearly explain this two-stage dataset preparation and training procedure.
>
> ---
> **Q: To clarify, would the training dataset preparation process require backpropagating through the base T2I model (and VAE)? Or does DISCO only need to do forward passes through them? (Addressing weakness 1 would help with clarifying this.).**
>
> A:  Our training dataset preparation **does not require backpropagation** through the base T2I model or the VAE. As noted earlier, our process is similar to VQGAN in that we only perform **forward passes** to extract the necessary representations. Backpropagation through the base model is **not involved** during this stage.
>
> ---
> **Q: What is the peak VRAM usage of training DISCO (vs. ControlNet, IP-Adapter, etc.)? That is often also an important bottleneck for training conditional generation models/adapters. Moreover, what is the exact speed comparison of DISCO vs. the baseline training-based methods? The RTX 3090 and A6000 are not exactly comparable regarding speed as they are not the same GPU.**
>
> A:  A key advantage of DISCO lies in its significantly reduced training cost—especially in terms of GPU memory and compute time. Traditional conditional generation methods such as ControlNet, IP-Adapter, and AnyControl require an auxiliary control network appended to the diffusion model. During training, the base diffusion model must still run in the forward pass and receive gradient updates in the backward pass, resulting in substantial computational overhead and memory consumption. These methods often require high-end GPUs with 48GB of memory.
>
> In contrast, DISCO avoids these costs by decoupling the conditional component from the diffusion model entirely. The transformer in DISCO is trained without executing the diffusion model in either the forward or backward pass, which enables much more efficient training.
>
> Since DISCO does not require loading the full Stable Diffusion (SD) model into GPU memory during training, it achieves substantial savings in both memory usage and training time. The table below summarizes the peak VRAM usage and total GPU hours, with all measurements conducted on the same hardware (NVIDIA A6000) to ensure a fair comparison:
>
> | Method    | ControlNet | T2I-Adapter | DISCO |
> | --------- | ---------- | ----------- | ----- |
> | GPU Hours | 500        | 290         | 65    |
> | Peak VRAM | 42 GB      | 31 GB       | 20 GB |
>
> These results demonstrate that DISCO achieves significant reductions in both memory usage and training time compared to existing conditional generation methods.
>
> ---
> **Q: The paper frames DISCO in terms of DDPM sampling, but will DISCO (like ControlNet and other training-based baselines) work with alternative/newer sampling methods like DPS and flow-matching trained models, especially since many SOTA T2I models are flow-matching models?**
>
> A: We appreciate the reviewer’s observation. This is indeed a valid concern, and we regret not including a discussion of this issue in the original submission due to space constraints.
>
> Recent works such as “Closing the ODE–SDE Gap in Score-Based Diffusion Models through the Fokker–Planck Equation” and “Flow-GRPO: Training Flow Matching Models via Online RL” propose effective techniques for converting deterministic ODE-based sampling processes into equivalent SDEs. These methods preserve the marginal distributions across time steps and enable reinterpretation of DDIM and Flow Matching trajectories as stochastic processes.
>
> By leveraging these ODE-to-SDE conversion techniques, our method—originally designed for stochastic DDPM samplers—can be naturally extended to support DDIM and Flow Matching frameworks as well.
>
> ---
> **Q: Since DISCO limits the selection of the guidance noise to a discrete codebook, would it potentially limit the diversity of the output images for higher resolution and more expressive T2I models (i.e., newer models after SD 1.5), especially considering training efficiency?**
>
> A:  This is an excellent question. Indeed, using a small codebook would limit the model's capacity and could result in poor image quality or even generation failure, let alone capturing diverse outputs.
>
> Based on our experiments, a codebook size of 4096 entries strikes a good balance between expressiveness and efficiency—it performs well in generating high-quality and diverse images, even with SD 1.5.
>
> However, for users seeking even higher fidelity or working with more expressive or higher-resolution models (e.g., SDXL or beyond), a larger codebook may be beneficial. That said, increasing the codebook size requires the transformer to have a larger output vocabulary, which in turn increases both training and inference costs—not just in terms of memory, but also latency and compute.
>
> In summary, while the discretization inherently introduces a trade-off between diversity and efficiency, our design allows flexibility: users can select an appropriate codebook size depending on the desired quality-performance balance.
>
> ---
> **Q:  DISCO is not evaluated with a user study, especially since FID and the condition-alignment metrics are often not entirely indicative of human preference.**
>
> A:  Thank you for your suggestion regarding the user study. To further validate the effectiveness of our method, we conducted a user study, the results of which are summarized in the table below. For each conditioning type, we randomly selected 50 images. The outputs from different methods were presented in random order to avoid positional bias. Participants were asked to choose the best result based on three criteria: realism, consistency with the text prompt, and consistency with the input condition.
>
> Each participant answered three formal questions per condition, plus one validation question, resulting in a total of 28 questions. Responses that failed the validation question were excluded from analysis. To compute the aggregate score, we applied a weighted scoring scheme across the three criteria: realism (30%), text alignment (30%), and condition alignment (40%).
>
> We invited volunteers from our research group of nearly 100 students to participate in the study. In total, we collected 76 questionnaires, of which 72 were considered valid after filtering out responses that failed the validation check.
>
> |                  | Depth | Canny |  HED | MLSD |  Seg | Normal | Skeleton | Location | Sketch |
> | ---------------- | :---: | :---: | :--: | :--: | :--: | :----: | :------: | :------: | :----: |
> | **UGD**          |   –   |   –   |   –  |   –  | 0.12 |    –   |     –    |     –    |    –   |
> | **FreeControl**  |  0.10 |  0.13 | 0.30 |   –  | 0.11 |  0.34  |   0.11   |     –    |    –   |
> | **ControlNet**   |  0.21 |  0.28 | 0.33 | 0.42 | 0.23 |  0.27  |   0.14   |   0.37   |  0.41  |
> | **T2I-Adapter**  |  0.22 |  0.11 |   –  |   –  |   –  |    –   |   0.25   |     –    |    –   |
> | **AnyControl**   |  0.18 |  0.17 |   –  |   –  | 0.24 |    –   |   0.19   |     –    |    –   |
> | **DISCO (Ours)** |  0.29 |  0.31 | 0.37 | 0.58 | 0.30 |  0.39  |   0.31   |   0.63   |  0.59  |
>
> The table reports the proportion of participants who preferred each method for a given condition. A dash (–) indicates that no official model was released for that conditioning type. As shown, our method consistently outperforms others across most conditions, demonstrating its superior controllability and perceptual quality in the eyes of human users.

---

### Note · Authors · 2025-08-14

**Dear Area Chair and Reviewers,**

We are truly grateful for the time and dedication you have devoted to evaluating our manuscript. Initially receiving scores of 5, 4, 4, and 3 (5443), our work was commended for its rationale, innovation, and experimental validation. During the rebuttal phase, we addressed each point of feedback in detail and were encouraged by the favorable responses from most reviewers.

It appears that we have resolved the concerns of Reviewers BpCV, RMAQ, and qShU, but not those of Reviewer gWpC. Reviewer gWpC noted that our method is not the first to use a codebook and questioned its novelty. They also raised the issue of a potential trade-off between representational capacity and efficiency for discrete noise, suggesting that covering enough style elements may require many codes and higher costs. They further expressed concern that Line 175 and Equation (7) might imply large computational overhead, and requested more analysis of training and inference costs since efficiency is a main claimed contribution.

Regarding novelty, we stress that our contribution is not the use of a codebook itself—a widely adopted technique—but the insight that a conditional control signal can be represented by a discrete codebook generated from random noise, combined with a transformer-based approach to generate such signals for conditional generation. This conditional generation framework is, to our knowledge, novel.

Regarding efficiency, “representational efficiency” here means that lower efficiency requires more codes to capture the control signal. We provide both a theoretical guarantee (Theorem 2) and experiments showing that a 4,096-entry codebook is sufficient. Equation (7) simply denotes selecting one code from the codebook, a standard VQGAN operation that does not add extra cost. The efficiency claimed in our work refers specifically to training efficiency. Conventional methods require loading the full diffusion model into GPU memory, incurring heavy costs. By discretizing the conditional signal, we are, to our knowledge, the first to train without loading the diffusion model, greatly reducing training time and memory usage.

Below, we outline the planned revisions to further strengthen the paper:

* Add user study and efficiency comparison to the paper.
* Provide a more detailed explanation of discrete noise prediction.
* Include additional details on training dataset preparation.

Best Wishes!

**The Authors**

---

### Decision · Program_Chairs · 2025-09-17

**Decision:**

Accept (poster)

**Comment:**

The paper introduces a novel technique for conditional generation in diffusion models based on a codebook, which allows for decoupling conditional training from the diffusion model entirely. Theoretical justifications are included and appreciated.

**Strengths**: Training without backpropagation through the diffusion model, which results in reduced GPU memory usage and training time (20GB vs. 48GB RAM, 65 GPU-hours vs. 290–500 GPU-hours in baselines).

**Weaknesses**: **Codebook novelty**: Codebooks are well-known techniques (e.g., VQVAE, DALLE). However,  their use for control conditioning is novel. **Scope**: Could be  generalized to text-only conditioning; not only **multimodal** settings. **LImited improvments**: Improvements on metrics like **SSIM** and **mIoU** are modest or inconsistent across baselines.


**Final justification:**  I find the method novel. While results are not significantly better, its training decoupling from diffusion models is a notable contribution, offering real-world benefits in terms of efficiency and scalability.

**Discussion summary:**
**Reviewer qShU**: Initially concerned about user study absence and unclear dataset prep. Post-rebuttal, new user study, and clarifications; maintained **Accept**. **Reviewer RMAQ**: Raised issues about experimental scope and efficiency evaluation. Authors clarified the multi-modal focus and presented comparative hardware-controlled efficiency data. Score increased from borderline to **Accept**. **Reviewer BpCV**: Appreciated flexibility and clarity but mentioned incompatibility with modern samplers. Authors addressed this issue in the ODE-SDE discussion, but the reviewer maintained the score at **Borderline Accept**. **Reviewer gWpC**: Maintained **Reject**, citing limited novelty, efficiency doubts, and weak metric gains. Authors responded with theoretical and empirical support, but this did not sway the reviewer.